# Reconciling Work and Family Demands and Related Psychosocial Risk and Support Factors among Working Families: A Finnish National Survey Study

**DOI:** 10.3390/ijerph19148566

**Published:** 2022-07-13

**Authors:** Janina M. Björk, Johanna Nordmyr, Anna K. Forsman

**Affiliations:** 1Department of Developmental Psychology, Faculty of Education and Welfare Studies, Åbo Akademi University, 65101 Vaasa, Finland; 2Health Sciences, Faculty of Education and Welfare Studies, Åbo Akademi University, 65101 Vaasa, Finland; johanna.nordmyr@abo.fi (J.N.); anna.k.forsman@abo.fi (A.K.F.)

**Keywords:** psychosocial support and risk factors, work–family conflict, gender equality, Finland, surveys and questionnaires, regression analysis

## Abstract

Working families commonly struggle with reconciling work and family demands. While the Nordic welfare states have been regarded as forerunners in family-friendly policies, worldwide trends threaten work–family reconciliation also in this context. Therefore, this study aimed to examine the associations between family interference with work (FIW)/work interference with family (WIF) and selected psychosocial risk and support factors in the work and family settings of Finnish working families. Data from the Finnish Quality of Work Life Survey 2018 collected by Statistics Finland were utilized to conduct binary logistic regression analyses (N = 1431). Risk factors in the work setting emerged as key covariates as all of them showed statistically significant associations with WIF or both WIF and FIW. Another key finding was that occasional conflicts within the family were beneficial in the context of both WIF and FIW. To conclude, both distinct and mutual psychosocial risk and support factors of FIW and WIF were identified, at the same time as two socio-demographic factors as well as one workplace factor were identified as covariates specifically of FIW. This study showed that work–family reconciliation is a considerable challenge among Finnish working families, and especially to women.

## 1. Introduction

There is a growing, multidisciplinary research interest in work–family reconciliation [1], and related concepts [2]. Due to worldwide contemporary trends, such as technological advancements, increased cross-national work, and the shift from single-career to dual-career couple households, working families are increasingly exposed to work–family conflict [3,4,5], which occurs when work and family demands conflict [6]. Most researchers argue that the conflict can be bidirectional [7] since evidence of family interference with work (FIW) and work interference with family (WIF) as related but distinct concepts is growing [8,9,10].

Work–family conflict is a public health concern demanding research attention due to its multiple outcomes [11], including individual-level mental and physical health problems [12,13], organizational-level absenteeism and turnover intentions [14], and societal-level healthcare costs [15]. While the potential consequences of work–family conflict are well-covered in previous research, less is known about its risk and support factors and their relative associations to FIW and WIF, although this body of literature is continuously growing [4,16,17,18].

Consistent with conflict theory [6], from which the concepts of WIF and FIW origin, it would be reasonable to expect that psychosocial family factors relate more to FIW than to WIF, while psychosocial work factors relate more to WIF than to FIW, and socio-demographic factors are equally related to both FIW and WIF since they may simultaneously influence both domains. The notion that work factors are more strongly associated with WIF than with FIW has repeatedly been supported in empirical research, e.g., [1,3,4]. For example, employees who perceive little support from co-workers and superiors report more WIF than FIW, and compared to family support (from family or other close ones in the family domain), work support (e.g., in terms of superior and co-worker support) is more strongly associated with WIF [3,4]. Further, employees who spend more time at work, and who experience task overload and psychological demands (e.g., a high work pace) tend to report more WIF than FIW [3]. In contrast, the empirical evidence on family factors’ stronger associations with FIW (as compared to WIF) is less consistent [1,3,4]. For example, while interpersonal conflicts within the family, support from family members and close ones overlap in their associations with FIW and WIF, the time individuals spend on family-related responsibilities and the role conflict they experience (i.e., the presence of competing, incompatible demands which require compromise) have been demonstrated to have stronger links to FIW than to WIF [3,4]. Relative relationship intensities aside, empirically driven studies seem to agree that cross-domain influences exist, suggesting that some work and family factors can influence the individuals’ family and work life at the same time [1,3,4,10,19].

While socio-demographic factors have not been identified as significant predictors of FIW/WIF, they influence the associations between psychosocial work and family factors and FIW/WIF [3,10,16]—supporting the use of social categories as covariates in such analyses [20].

To address work–family conflict issues, welfare states have implemented various family-friendly policies, with the Nordic countries positively standing out in international comparisons [21,22]. A characteristic of the Nordic welfare states is the well-established cooperation system existing between the government, employers’ organizations, and trade unions [5]. For example, the Nordic countries have been recognized for high-quality publicly funded childcare services, shared and paid parental leave, and flexible work arrangements for parents [5,21,23], resulting in low levels of work–family conflict [21,22]. At the same time, some studies report contradicting findings [24,25], and the Nordic welfare model is increasingly challenged by societal changes as well as criticized for not responding to them [5,23,26]. Taken together, family-friendly policies which have been designed and implemented by communities and work organizations may no longer correspond to contemporary work and family life, and this increasingly applies to the Nordic welfare states as well, warranting studies on the social circumstances and related psychosocial risk and support factors of certain population groups.

From a public health perspective, it is important to acknowledge that societal trends affect population groups differently, exposing them to varying levels of work–family conflict. Research on which particular factors support and hinder work–family reconciliation for couples has been called for, including studies targeting the Nordic context [26]. Romantic relationships are complex, as they have been demonstrated to be associated with enhanced well-being [27] but also stress [28], and in line with often adopted family systems theory [29], it can be argued that family members’ demands from work and family are interrelated with each other’s working conditions [26]. At the same time, a common assumption in international research is that children intensify the work–family conflict of working families due to increased family demands [1]. Support for this assumption has been found, especially regarding working mothers, since they experience high levels of parental demands [30,31,32], and working families with young children, since these are the most time-pressed—they simultaneously must earn money and provide childcare [32,33,34]. Considering that a relatively large proportion of the world population is living in a family with children, in Finland this frequency was 37% in 2020 [35], work–family conflict among working families therefore requires closer attention in research.

Taken together, work–family conflict is increasingly considered a public health concern also in the Nordic countries given that the previously tributed welfare model is now subject to a growing criticism as it fails to meet the needs related to contemporary societal trends. More research is needed, examining what psychosocial factors support and hinder successful work–family reconciliation for population groups with varying social circumstances and related prerequisites, so that future public policies can better address their needs and expectations. Therefore, the aim of this study was to examine the associations between FIW/WIF and selected psychosocial risk and support factors in the work and family settings of Finnish working families. Since disproportionate focus in previous research has been directed to those experiencing interference [1,18], those reporting no interference were in focus in the current study. The bidirectionality in interference was highlighted (WIF and FIW), comparing those who experience no interference with those who experience interference between work and family.

## 2. Materials and Methods

### 2.1. Study Design and Data Material

The current study was based on national interview survey data from the Finnish Quality of Work Life Survey 2018 (QWLS) collected by Statistics Finland, a governmental national statistics service provider. The study targeted participants aged 15–67 who were identified as employed wage and salary earners regularly working at least 10 h per week. The interviews were primarily conducted face-to-face (9% were conducted over the phone), and the duration median of the face-to-face interviews was 63 min. The number of persons participating in the QWLS was 4110, giving a response rate of 66.8% [36]. Given the aim of the current study, the inclusion criteria specified that respondents had to live in a household with children under 18 years and be involved in a cohabiting relationship (i.e., married, engaged, or registered partnership). The final number of participants in our study sample was 1431.

### 2.2. Measures

Two directions of work–family conflict (FIW and WIF) were measured using single-item statements. The dichotomization of these two dependent variables was in line with the study aim, focusing on how the group of respondents reporting no interference distinguished from the group of respondents reporting any or significant interference.

Further, four socio-demographic (chronological age, gender, educational level, and age of children living in the household) and four workplace (temporal work flexibility, spatial work flexibility, employment type, and number of subordinates) characteristics were included in the analysis. The original, dichotomous categorization was kept for gender, temporal work flexibility, and employment type, while the rest of these variables were recoded.

Based on previous empirical research, e.g., [1,3,4], selected psychosocial work and family factors were also included in the analysis. Namely, three risk (overtime, task overload, and work pace) and two support factors (superior support and co-worker support) in the work setting, and three risk (only part-time work, task reduction, and refused more work demands) and two support (family support and support from close ones) factors in the family setting. All risk factors in the work setting as well as the family factor support from close ones were initially scored on Likert-scales and recoded into dichotomous variables, while the original categorization was kept for all other work and family factors.

The recoding process (including original and recoded variables, survey items, and response options) is presented in detail in Appendix A.

### 2.3. Statistical Analysis

SPSS version 27 was used to conduct the statistical analyzes. A missing data analysis revealed that the missing values ranged from 0 to 3 (0.002%) for the included variables. The responses ‘not applicable’ and ‘cannot say’ ranged from 4 to 80 (0.3–5.6%) and from 0 to 3 (0–0.2%) respectively. Descriptive statistics were used to report sample characteristics (i.e., frequencies and percentages).

Next, the Pearson’s chi-square test was used to conduct between-group comparison of reported WIF and FIW in relation to the included variables. This was followed by binary logistic regression analyzes with reported FIW and WIF as the dependent variables. The regression analyses were conducted manually and stepwise by entering the dependent variables, socio-demographic, and workplace characteristics in step 1, and by adding the psychosocial work and family factors in steps 2 and 3, respectively. The logistic regression analyses were conducted using the Enter method. The results are presented in terms of calculated odds ratios with 95% confidence intervals. The models’ goodness of fit is estimated by Hosmer and Lemeshow goodness-of-fit test.

## 3. Results

### 3.1. Descriptive Statistics

Study sample characteristics are presented in Table 1 (work–family conflict, socio-demographic, and workplace characteristics) and Appendix A (psychosocial work and family factors). With regards to the socio-demographic characteristics, the study sample (*N* = 1431) consisted of 741 (51.8%) women and 690 (48.2%) men, respondents aged 35–44 represented the largest age group (46.1%), while respondents aged 55–67 represented the smallest (4.2%), and there was an even distribution between low (52.4%) and high (47.6%) educational level. The socio-demographic and workplace characteristics of the current sample were distributed in similar ways as in the total QWLS-sample (*N* = 4110). Further, all correlations between variables included in the model were below 0.70 (*p* < 0.05). This revealed no signs of significant multicollinearity problems, which correlations above 0.80 tend to indicate [37].

Moreover, 31.9% of the respondents in our study sample reported no FIW, and 26.6% no WIF. Table 2 presents the distribution (%) and between-group comparison of socio-demographic and workplace characteristics among participants according to reported FIW/WIF status, and Appendix A shows the distribution of perceived psychosocial risk and support factors in the work and family settings.

### 3.2. The Association between Perceived Psychosocial Risk and Support Factors in the Work and Family Settings and FIW/WIF

The main results of the regression analyses remained stable across models. Therefore, only the final model (i.e., step 3) for both dependent variables (FIW and WIF) is presented in Table 3 as well as in the running text. Regarding FIW, three of the socio-demographic and workplace characteristics were statistically significant. That is, the odds for reporting no FIW were lower for respondents aged 35–44 than for respondents aged 20–34. Further, men and non-teleworkers were more likely to report no FIW than women and teleworkers.

Moreover, we found that two of the examined psychosocial risk and support factors in the work setting, task overload and superior support, were associated with FIW. Specifically, regarding task overload, respondents perceiving more task overload were significantly less likely to report no FIW compared to those perceiving less task overload. Regarding perceived superior support, respondents responding ‘Often’ were significantly less likely to report no FIW than those responding ‘Never’. However, those responding ‘Sometimes’ or ‘Always’ did not statistically differ from those responding ‘Never’.

Additionally, the odds for no FIW were higher for respondents who perceived occasional family conflict compared to those who perceived frequent family conflicts. However, respondents who no longer have or ever had perceived family conflicts did not differ statistically from the group that perceived frequent family conflicts. This was the only variable showing statistically significant associations with FIW among the psychosocial family factors.

Regarding WIF, the results demonstrate that none of the socio-demographic and workplace characteristics significantly predicted WIF.

Further, the statistical analysis showed that all perceived risk factors, but no support factors, in the work setting had statistically significant associations with WIF. That is, respondents who perceived low work risks were also more likely to report no WIF than those who perceived high risks.

The results show that both a risk and a support factor were statistically significant in the family setting. Specifically, the respondent groups which perceived that they had never had to reduce work tasks due to family reasons and occasional family conflicts had a higher probability for reporting no WIF than their respective reference groups.

## 4. Discussion

In this study, approximately a third and a quarter of the respondents reported no FIW and no WIF, respectively, demonstrating that even though a fair share of the respondents successfully had reconciled demands from work and family, reconciliation was still a considerable challenge to the majority of respondents.

By comparing the group reporting no FIW/WIF with the group reporting FIW/WIF, statistically significant differences were found with regards to all examined risk factors at work. Specifically, perceiving no or low task overload was associated with no interference in both directions. Overtime- and work pace-variables were similarly associated with WIF but not with FIW. Consistent with previous meta-analytical findings, our findings support the notion that a stressful and time-demanding work hinders work–family reconciliation, and that risk factors in the work setting are more frequently related to WIF than to FIW e.g., [3].

Moreover, while previous research has repeatedly emphasized that various kinds of social support at work reduce WIF and FIW alike [3,4], our study findings demonstrate less relevance of social support at work and the results are in part contradicting. Namely, superior support was the only work support variable showing statistically significant associations with FIW—those who often perceived superior support were less likely to report no FIW than those who never perceived superior support. However, this result should be interpreted with caution since we found no such systematic differences between the rest of the respondent groups with regards to FIW, and no systematic differences were found between any of the respondent groups with regards to WIF. Perhaps, the benefit of social support from managers or co-workers is restricted to specific situations when it is needed or perceived as useful by working families. Instead, it may be that broad, organizational support (e.g., family-friendly organizational policies, attitudes, and behaviors), more effectively supports work–family reconciliation [4].

Regarding the role of family factors for WIF/FIW, reducing job tasks due to family responsibilities was a risk factor with regards to WIF in our study sample. While it might seem more logical that statistically significant associations would have been found between this variable and FIW, cross-domain influences have been found in previous meta-analytic research as well [1,10]. Further, we speculate that a coping strategy of those who had not reduced job tasks has been to consciously choose less demanding job roles during child-rearing years. Therefore, we call for studies investigating how individuals navigate work and family during different life-stages.

Moreover, we found that the variable measuring family support was significantly associated with both directions of interference. Previous research findings have suggested that family conflicts increase both FIW and WIF [3]. While our results indeed highlight that frequent conflicts might drain working families, they also suggest that occasional conflicts might be the right middle ground, being vital to an open communication climate and reducing interpersonal stress within working families [28].

Even though this study did not particularly focus on socio-demographic and workplace characteristics, it was interesting to note that age, gender, and spatial flexibility were statistically significant covariates of FIW (none of WIF) in the current sample.

We included children’s age as a covariate since previous studies have shown that especially young children might amplify the work–family conflict through increased family demands [32,33,34]. However, this variable proved non-significant in the current study. This finding may suggest that the governmental support offered to working families in a Finnish context is more useful to families with young children, thereby diminishing differences between them and families with older children.

Finally, no FIW was more common among men than women in this sample of working families. A recent meta-analysis focusing on gender differences in work–family conflict reported similar results—while the gender effects tended to be small, among the more significant gender effects was mothers reporting greater FIW than fathers [16]. Further, our finding stresses that even though gender equality in many areas (e.g., education, employment, and health) is supported by the Finnish government, women still experience gender inequality in relation to work–family reconciliation. In line with our results, two recent, large-scale, comparative studies have shown that the level of gender equality in the society is an important factor to consider in work–family conflict research. High levels of gender equality in society combined with individual-level egalitarian values are, for example, associated with higher levels of burnout among mothers [30], and while living in a society characterized by gender equality reduces work–family conflict, it also strengthens the negative relationship between work–family conflict and well-being [31]. Thus, the governmental support of gender equality in other areas may have a rather paradoxical effect when inequalities are still existing in parenting. This points to the urgency of promoting gender equality in the family setting for countries, such as Finland, which generally are viewed as forerunners in terms of gender equality and related policy and practice development. Thus, we call for research emphasizing the female perspective on work–family conflict in various national contexts, at the same time as we highlight the emerging issues in the Nordic welfare state setting.

### 4.1. Strengths and Limitations

This study was based on data from the Finnish QWLS, a national survey study with a relatively high response rate (66.8%). The sample characteristics were well representative of the total study population. Further, interviews were primarily conducted face-to-face in this large-scale, high-quality survey, which could be regarded a strength [36]. However, the cross-sectional design means that no causal interferences could be determined and there was a risk of common method bias.

The use of binary logistic regression is associated with both strengths and limitations. Specifically, binary regression allows for studying groupwise differences while controlling for potential covariates, but dependent variables must be dichotomous, meaning that nuances of the data might remain undiscovered. However, we wanted to dichotomize these variables to separate the respondent group reporting no FIW/WIF from the groups reporting FIW/WIF, to identify systematic differences in what psychosocial risk and support factors they perceived.

Regarding the measurement of work–family conflict, we did not use a comprehensive scale. While this limits the present study, two important aspects of work–family conflict were indeed captured by conducting separate analyses for FIW and WIF [7], allowing us to distinguish both their mutual and distinct risk and support factors [8,9]. Also, single-item questions might be easy to grasp in an otherwise comprehensive survey from the viewpoint of respondents.

Finally, the main results turned out to be stable across models, but the stepwise process was necessary to conduct to reveal this pattern. Hence, the inclusion of socio-demographic and workplace characteristics as covariates of FIW/WIF may be regarded as strengthening the validity of the main findings related to the selected psychosocial work and family factors and their associations with FIW/WIF.

### 4.2. Implications for Research and Practice

Further studies should investigate how this relatively large population group can reconcile work and family, so that measures in work settings can be taken based on a solid evidence base. Here, studies adopting a lifespan approach and critical gender equality perspective are particularly warranted. Further, even though the governmental support in Finland is generally considered generous, several psychosocial risk factors in the work and family settings of working families were identified in this study. To remain effective, this implies that family-friendly public policy work must be iterative, critically and systematically evaluating perceived risk and support factors of working families.

## 5. Conclusions

Work–family conflict is a public health concern increasingly demanding attention also in the Nordic welfare context. Taking into consideration what psychosocial factors support and hinder successful work–family reconciliation for a vast population group with varying social circumstances and related prerequisites is necessary to properly address specific needs and expectations among the working age population.

Importantly, the current study highlights that work–family conflict is bidirectional. Examining psychosocial risk and support factors in the work and family settings of Finnish working families, risk factors in the work setting emerged as especially important covariates since all of them showed statistically significant associations with WIF or both WIF and FIW. In addition, occasional conflicts within the family proved beneficial in the context of both WIF and FIW. To conclude, both distinct and mutual psychosocial risk and support factors of FIW and WIF were identified, at the same time as two socio-demographic factors as well as one workplace factor were identified as covariates of FIW. This study contributes to the literature on work–family conflict by showing that reconciling work and family is a considerable challenge to Finnish working families despite the governmental support offered in this welfare state—especially to women.

## Figures and Tables

**Table 1 ijerph-19-08566-t001:** Overview of the study sample according to variables measuring work–family conflict and socio-demographic and workplace characteristics. *N* = 1431.

Variable	Response Category	*N* (%)
*Work–family conflict*		
Family interference with work (FIW)	Reported FIW	963 (67.4)
	Reported no FIW	452 (31.6)
	N/A	15 (1.0)
Work interference with family (WIF)	WIF	1045 (73.1)
	No WIF	378 (26.4)
	N/A	7 (0.5)
*Socio-demographic and workplace characteristics*		
Age	20–34	284 (19.8)
	35–44	659 (46.1)
	45–54	428 (29.9)
	55–67	60 (4.2)
Gender	Woman	690 (48.2)
	Man	741 (51.8)
Educational level	Low	750 (52.4)
	High	681 (47.6)
Temporal flexibility	Fixed	431 (30.1)
	Flexible	1000 (69.9)
Spatial flexibility	No telework	934 (65.3)
	Telework	497 (34.7)
Employment type	Full-time	1312 (91.7)
	Part-time	116 (8.1)
Number of subordinates	No subordinates	1027 (71.8)
	1–9	233 (16.3)
	10 or more	169 (11.8)
Age of children	0–7 years only	420 (29.4)
	8–17 years only	712 (49.8)
	Mixed	299 (20.9)

Missing data ranged from 0 (0%) to 3 (0.002%) for the included variables. N/A = Not applicable. After initial, descriptive analyses, ‘not applicable-’, and ‘cannot say-’ responses were excluded.

**Table 2 ijerph-19-08566-t002:** The distribution and between-group comparison of socio-demographic and workplace characteristics among participants according to reported family interference with work (FIW)/work interference with family (WIF) status. *N* = 1431.

	FIW (%)	No FIW (%)	χ^2^	WIF (%)	No WIF (%)	χ^2^
*Socio-demographic and workplace characteristics*						
Age			*p* ≤ 0.001			*p* = 0.076
20–34	178 (63.1)	104 (36.9)		204 (72.1)	79 (27.9)	
35–44	491 (75.1)	163 (24.9)		501 (76.4)	155 (23.6)	
45–54	267 (63.6)	153 (36.4)		302 (71.1)	123 (28.9)	
55–67	27 (45.8)	32 (54.2)		38 (64.4)	21 (35.6)	
Gender			*p* ≤ 0.001			*p* = 0.051
Woman	502 (73.5)	181 (26.5)		520 (75.8)	166 (24.2)	
Man	461 (63)	271 (37)		525 (71.2)	212 (28.8)	
Educational level			*p* ≤ 0.001			*p* ≤ 0.001
Low	463 (62.6)	277 (37.4)		509 (68.4)	235 (31.6)	
High	500 (74.1)	175 (25.9)		536 (78.9)	143 (21.1)	
Temporal flexibility			*p* = 0.345			*p* = 0.237
Fixed	283 (66.3)	144 (33.7)		306 (71.3)	123 (28.7)	
Flexible	680 (68.8)	308 (31.2)		739 (74.3)	255 (25.7)	
Spatial flexibility			*p* ≤ 0.001			*p* ≤ 0.001
No telework	592 (64.2)	330 (35.8)		645 (69.4)	285 (30.6)	
Telework	371 (75.3)	122 (24.7)		400 (81.1)	93 (18.9)	
Employment type			*p* = 0.537			*p* = 0.356
Full-time	880 (67.9)	416 (32.1)		962 (73.8)	342 (26.2)	
Part-time	82 (70.7)	34 (29.3)		81 (69.8)	35 (30.2)	
Number of subordinates			*p* = 0.196			*p* = 0.006
No subordinates	688 (67.6)	329 (32.4)		728 (71.2)	295 (28.8)	
1–9	167 (72.6)	63 (27.4)		185 (80.1)	46 (19.9)	
10 or more	107 (64.5)	59 (35.5)		131 (78.4)	36 (21.6)	
Age of children			*p* = 0.014			*p* = 0.018
0–7 years only	287 (69)	129 (31)		315 (75.4)	103 (24.6)	
8–17 years only	455 (64.9)	246 (35.1)		496 (70.3)	210 (29.7)	
Mixed	221 (74.2)	77 (25.8)		234 (78.3)	65 (21.7)	

**Table 3 ijerph-19-08566-t003:** Odds ratio with 95% confidence intervals of reporting no family interference with work (FIW)/no work interference with family (WIF).

All *N* = 1431			
		FIW	WIF
Age	20–34	**1.00**	1.00
	35–44	**0.68 (0.46–0.99)**	0.84 (0.56–1.26)
	45–54	1.09 (0.68–1.73)	0.99 (0.60–1.63)
	55–67	1.23 (0.58–2.60)	0.71 (0.32–1.59)
Gender	Woman	**1.00**	1.00
	Man	**1.53 (1.12–2.09)**	1.31 (0.94–1.83)
Educational level	Low	1.00	1.00
	High	0.81 (0.61–1.08)	0.95 (0.69–1.29)
Temporal flexibility	Fixed	1.00	1.00
	Flexible	0.92 (0.68–1.25)	0.95 (0.68–1.31)
Spatial flexibility	No telework	**1.00**	1.00
	Telework	**0.71 (0.52–0.98)**	0.74 (0.52–1.04)
Employment type	Full-time	1.00	1.00
	Part-time	0.97 (0.56–1.65)	1.14 (0.66–1.96)
Number of subordinates	No subordinates	1.00	1.00
	1–9	0.87 (0.59–1.29)	0.84 (0.55–1.28)
	10 or more	1.27 (0.84–1.93)	0.88 (0.55–1.43)
Age of children	0–7 years only	1.00	1.00
	8–17 years only	1.02 (0.69–1.51)	1.30 (0.86–1.96)
	Mixed	0.83 (0.56–1.23)	0.81 (0.53–1-24)
Overtime	Agree	1.00	**1.00**
	Disagree	1.19 (0.88–1.60)	**1.87 (1.35–2.58)**
Task overload	Agree	**1.00**	**1.00**
	Disagree	**1.46 (1.08–1.98)**	**2.01 (1.47–2.76)**
Work pace	Agree	1.00	**1.00**
	Disagree	1.14 (0.84–1.56)	**1.41 (1.01–1.98)**
Superior support	Never	**1.00**	1.00
	Sometimes	0.96 (0.53–1.74)	0.80 (0.42–1.50)
	Often	**0.53 (0.30–0.91)**	0.60 (0.34–1.08)
	Always	0.73 (0.42–1.25)	0.77 (0.43–1.38)
Co-worker support	Never	1.00	1.00
	Sometimes	1.05 (0.41–2.66)	1.33 (0.48–3.73)
	Often	1.06 (0.43–2.64)	1.34 (0.49–3.67)
	Always	1.01 (0.40–2.57)	1.36 (0.49–3.82)
Only part-time work	Yes	1.00	1.00
	No	1.09 (0.77–1.55)	0.72 (0.50–1.03)
Task reduction	Yes	1.00	**1.00**
	No	1.20 (0.87–1.64)	**1.50 (1.07–2.10)**
Refused more work demands	Yes	1.00	1.00
	No	1.02 (0.73–1.44)	1.44 (0.99–2.11)
Family support	Frequent conflicts	**1.00**	**1.00**
	Occasional conflicts	**6.32 (2.58–15.45)**	**2.44 (1.13–5.25)**
	No conflicts anymore	3.66 (0.81–16.68)	0.29 (0.03–2.67)
	No conflicts	2.39 (0.98–5.82)	1.09 (0.51–2.33)
Support from close ones	Disagree	1.00	1.00
	Agree	1.17 (0.88–1.54)	0.98 (0.73–1.31)
Hosmer and Lemeshowgoodness-of-fit test		χ^2^ = 7.125, df = 8, *p* = 0.523	χ^2^ = 9.700, df = 8, *p* = 0.287

Statistically significant odds ratios (95% confidence intervals) in bold print.

## Data Availability

The release of Finnish Quality of Work Life Survey 2018 data is subject to a user license. Information on licenses is provided by Statistics Finland: https://www.tilastokeskus.fi/tup/mikroaineistot/hakumenettely_en.html (accessed on 13 June 2022).

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
