# Peer review of "Reconciling Work and Family Demands and Related Psychosocial Risk and Support Factors among Working Families: A Finnish National Survey Study"

_ijerph, 2022, doi:10.3390/ijerph19148566_

Round 1
Reviewer 1 Report
This paper takes up a relevant topic for workers worldwide, the two-way balance between work and family. Cross-sectional data from a national survey (N=1431) was analysed by binary logistic regression models including numerous independent variables and control factors. The authors conclude based on their exploratory analyses that distinct and mutual risk factors of FIW and WIF can be found. Please, find my comments and suggestions point by point below.
Abstract:
1. After having read the presentation of findings, I do not have a clear picture of what was the most important finding and the contribution of the study. I suggest the authors to consider ”painting with a broader brush” here – what is the main finding in general terms and what is the conclusion? As it reads now it looks like the conclusion is that society and workplaces needs to sharpen their efforts, but this is not what has been researched, rather a suggested implication of the study. I also suggest the authors to skip presenting suggestions for future studies in the abstract. In my oppinion, it would give the reader a better overview to focus on the contribution of the current study.
Introduction:
In general, I find the introduction has a good flow and presents the context of the Nordic welfare systems in an interesting way. Though, I have some suggestions for further improvement.
2. I miss a brief literature overview of previously research on risk and protective factors in relation to the two outcomes under study (By the way, the abbreviations WIF and FIW need to be spelled out the first time they are used).
3. P 2, L 80-89: From the previous text it is not clear what “more research is needed”. By adding a brief literature overview, I believe the positioning of the paper could be more clear. The same applies to the aim, which is rather vague and exploratory and would benefit for being sharpened.This would probably also be a help to guide the revision of the other sections of the manuscript.
Methods:
4. A design question: why control for sociodemographic factors instead of testing hypotheses/presenting expectations related to one or more of these? In other words: what do we already know and what do we not know regarding work/family factors and demographics in relation to WIF & FIW.
5) P 3, l101-103: civil status and children in the home are inclusion criteria. I find the formulation “automatically control for” a bit weird in the context.
6) Table 1: I suggest the authors to summarize this briefly in the running text and move the table to an e-appendix for those interested in details.
7) Without the suggested literature overview (please, see comment 2) it looks like a “fishing trip” where everything that might be of the sligthest interest in the national survey was included in the study. Why are these factors relevant to include – and why are the same variables relevant for both outcomes?
8) section 2.3 statistical analysis (p 5 ): To my best knowledge, it is not a stepwise regression but a hierarchical block regression. I suggest the authors clarify why they chose this approach – I do not see it is utilised in the discussion. Maybe it would be sufficient to only present Model 3? Also, please provide further details regarding the regressions (estimation method, multicollinearity etc).
9) I suggest the authors to delete the reference to tables 2-4 in the running text here and save it to the results (i.e. “see Table 2”).
Results
10) Table 2 is far too detailed as it is. A suggestion could be to only present the work familiy conflict and demographics/workplace characteristics in Table 2 and the rest in an E-appendix. The same applies to Table 3. I understand that this is a first step for the researchers to get to know the data, but this level of details is far too much and with a following regression model bivariate analyses are less interesting. My suggestion is only to present what is highly relevant for the reader to know and skip the rest. – And a minor comment: no need to repeate Pearsons chi-square test (l151 p 7) etc. This has been specified in the methods section.
11) P 9, l 159: “Table 4 indicates that…” Please, reformulate to “the results presented in Table 4 show….” As ststistical significant findings is a solid finding and not an indication.
12) With such a detailed presentation of the regression results from 3 models for 2 outcomes I as a reader loose the overview. As the main pattern is stable across models it would be enough to mention that the models were initially run this way, but as the main results were stable across models only the final model is presented. Then 4 and 5 could be collapsed into one model only and the main findings specified across models. Still, I am sceptical to 1.5 pages tables and suggest the authors reconsider the exploratory approach and instead be more clear about specification of the models. It is a fundamental basics of regressions not to overload the models, but be cautious of specification.
Discussion
In general, I find the discussion works better than the results section. However, parts of the text is too close to results ( i.e. phrases such as ” those who responded “no” ..more frequently reported”. A lot of the information that is taken up in the discussion could be utilised to strengthen the background/rationale for the study. For example p 13, line 240-241 regarding childrens age.
Conclusion
Only a few sentences actually build on the findings while most is practical implications. The results do not warrant that family-friendly public policy work must be iterative etc. The conclusion needs to be based directly on the findings of the actual study. A solution could be to add a brief section of implications for research and practice just before the conclusion.
Author Response
Please see attachment if you prefer authors' responses in PDF-format.
Authors’ responses to reviewer comments: reviewer 1
Dear Reviewer,
We are very thankful for the valuable comments provided. Please see amendments marked with the track changes function in the submitted manuscript file. Please also see our point-by-point response to your comments below.
Comments (general):
This paper takes up a relevant topic for workers worldwide, the two-way balance between work and family. Cross-sectional data from a national survey (N=1431) was analysed by binary logistic regression models including numerous independent variables and control factors. The authors conclude based on their exploratory analyses that distinct and mutual risk factors of FIW and WIF can be found. Please, find my comments and suggestions point by point below.
Authors’ response:
We are happy to see that you share our view regarding the topic being relevant. Your summary of the study, as well as the related comments and suggestions demonstrate that you have a good understanding of the field and the applied statistical methods, as well as what this study sets out to do. Thus, we very much appreciate your expertise and how your input has helped us improve the manuscript.
Comments (abstract):
- After having read the presentation of findings, I do not have a clear picture of what was the most important finding and the contribution of the study. I suggest the authors to consider ”painting with a broader brush” here – what is the main finding in general terms and what is the conclusion? As it reads now it looks like the conclusion is that society and workplaces needs to sharpen their efforts, but this is not what has been researched, rather a suggested implication of the study. I also suggest the authors to skip presenting suggestions for future studies in the abstract. In my oppinion, it would give the reader a better overview to focus on the contribution of the current study.
Authors’ response:
Thank you for this remark. We have revised according to your suggestions, now focusing on the contribution of the current study by presenting the main findings in more general terms, adding a concluding remark to the abstract, and excluding suggestions for future studies.
Comments (introduction):
In general, I find the introduction has a good flow and presents the context of the Nordic welfare systems in an interesting way. Though, I have some suggestions for further improvement.
- I miss a brief literature overview of previously research on risk and protective factors in relation to the two outcomes under study (By the way, the abbreviations WIF and FIW need to be spelled out the first time they are used).
- P 2, L 80-89: From the previous text it is not clear what “more research is needed”. By adding a brief literature overview, I believe the positioning of the paper could be more clear. The same applies to the aim, which is rather vague and exploratory and would benefit for being sharpened.This would probably also be a help to guide the revision of the other sections of the manuscript.
Authors’ response:
We are happy to see that you in general find the introduction to have a good flow and that you consider the study context to be presented in an interesting way.
- We agree that an added brief overview of risk and protective factors that have been demonstrated to be associated with the two outcomes under study in previous research strengthens the current manuscript. Please see tracked changes on page 2, lines 50–82, for a more detailed summary of previous review and meta-analysis results. Please also note that when the abbreviations FIW and WIF are used in the abstract, main text, and the tables of the revised manuscript, they are spelled out, as recommended.
- It is our interpretation that you refer to the same literature overview here as in comment 2. Since we have followed your suggestion related to comment 2. and added a brief literature overview of the risk and protective factors in relation to the two outcomes under study, we see how this addition has helped us to clarify the positioning of the paper in the revised version of the manuscript. We have also clarified what we refer to in the sentence containing the phrasing ‘more research is needed’ (page 3, line 118), highlighting that further investigation is needed regarding what psychosocial factors support and hinder successful work-family reconciliation for population groups with varying social circumstances and related prerequisites. As recommended, we have specified the study aim (page 3 lines 121–124): “[…] the aim of this study was to examine the associations between FIW/WIF and selected psychosocial risk and support factors in the work and family settings of Finnish working families”, highlighting the main factors and related associations in focus.
Comments (methods):
- A design question: why control for sociodemographic factors instead of testing hypotheses/presenting expectations related to one or more of these? In other words: what do we already know and what do we not know regarding work/family factors and demographics in relation to WIF & FIW.
5) P 3, l101-103: civil status and children in the home are inclusion criteria. I find the formulation “automatically control for” a bit weird in the context.
6) Table 1: I suggest the authors to summarize this briefly in the running text and move the table to an e-appendix for those interested in details.
7) Without the suggested literature overview (please, see comment 2) it looks like a “fishing trip” where everything that might be of the sligthest interest in the national survey was included in the study. Why are these factors relevant to include – and why are the same variables relevant for both outcomes?
8) section 2.3 statistical analysis (p 5 ): To my best knowledge, it is not a stepwise regression but a hierarchical block regression. I suggest the authors clarify why they chose this approach – I do not see it is utilised in the discussion. Maybe it would be sufficient to only present Model 3? Also, please provide further details regarding the regressions (estimation method, multicollinearity etc).
9) I suggest the authors to delete the reference to tables 2-4 in the running text here and save it to the results (i.e. “see Table 2”).
Authors’ response:
We very much appreciate your comments relating to the methods.
- After considering and reflecting upon this comment, we agree that demographics should be presented as covariates (independent variables) of FIW and WIF (and not just as control variables) in the current study. In the revised manuscript, we therefore present them as such to acknowledge that they are important to consider in relation to work-family conflict. However, since we also agree with you regarding the need to sharpen the study focus and clarify the positioning of the paper (comment 3.), we have clarified that this study is empirically driven, and as such, our focus is on the selected psychosocial factors. Previous research demonstrates that psychosocial factors in the work and family settings are more strongly related to FIW/WIF than socio-demographic characteristics, but socio-demographic characteristics have been found to influence associations between work/family factors and FIW/WIF.
- We have revised by stating that civil status and having children in the home are sample inclusion criteria and removed the “automatically control for”-formulation (page 3, lines 139–142).
- Table 1 now constitutes a supplementary table (Table S1), and the recoding process is merely briefly described in the running text.
- In the revised and developed literature overview that we have amended based on your suggestion, we believe that it now becomes clear to the reader why we have chosen to focus on the included variables. The included variables were selected based on previous empirical research, and the ambition was to attain a good balance between work- and family-related factors in the analyses. The underlying reason to why we tested the same variables in both regression analyses was that FIW and WIF have been found to be related but distinct concepts, with cross-domain associations demonstrated in previous research (stated on page 2, lines 45 and lines 71–74). Please also see the added overview of risk and protective factors [page 2, lines 50–82].
- After considering your comments relating to the statistical analysis and presentation of findings, we have chosen to follow your suggestion and only present the final model (i.e. Model 3) for both outcomes in the revised manuscript. The main results turned out to be stable across models, but the stepwise process was regarded necessary to conduct to reveal this pattern. This is now discussed in 4.1 Strengths and limitations (page 17, lines 378–382). Regarding the stepwise logistic regression analysis, we refer to the analytical process where we entered the variables manually and stepwise using the Enter method (see page 6, lines 184–188). Further, as suggested we now provide details on the analysis revealing no signs of multicollinearity problems (see page 9, lines 205–207).
Comments (results):
10) Table 2 is far too detailed as it is. A suggestion could be to only present the work familiy conflict and demographics/workplace characteristics in Table 2 and the rest in an E-appendix. The same applies to Table 3. I understand that this is a first step for the researchers to get to know the data, but this level of details is far too much and with a following regression model bivariate analyses are less interesting. My suggestion is only to present what is highly relevant for the reader to know and skip the rest. – And a minor comment: no need to repeate Pearsons chi-square test (l151 p 7) etc. This has been specified in the methods section.
11) P 9, l 159: “Table 4 indicates that…” Please, reformulate to “the results presented in Table 4 show….” As ststistical significant findings is a solid finding and not an indication.
12) With such a detailed presentation of the regression results from 3 models for 2 outcomes I as a reader loose the overview. As the main pattern is stable across models it would be enough to mention that the models were initially run this way, but as the main results were stable across models only the final model is presented. Then 4 and 5 could be collapsed into one model only and the main findings specified across models. Still, I am sceptical to 1.5 pages tables and suggest the authors reconsider the exploratory approach and instead be more clear about specification of the models. It is a fundamental basics of regressions not to overload the models, but be cautious of specification.
Authors’ response:
- Thank you for these observations. We can see how less detailed tables (previously Table 2 and 3) in which only the most important information is provided increases readability and directs attention to the main findings. Following your suggestion, we have shortened the tables by only presenting FIW, WIF, and socio-demographic and workplace characteristics in these tables (currently Table 1 and Table 2) and the rest in supplementary tables Table S2 and Table S3. Also, we have removed the first paragraph from 3.2 in which we presented the results of the Pearson’s chi-square test (in the revised manuscript these results are only presented in Table 2 as well as in Table S3) to highlight the main findings of the study.
- Thank you for noticing this. We have revised accordingly.
- This is an important comment. We agree, it is enough to mention in the running text that the models were initially run this way, but as the main results were stable across models only the final model for each outcome is presented. We have revised accordingly. You continue by commenting that: “Then 4 and 5 could be collapsed into one model only and the main findings specified across models.”. We interpret this part of the comment to refer to Table 4 and 5 and that these could be collapsed into a single table, presenting the findings of the final models for both outcomes. The results of the final regression models are now presented in a single table, Table 3.
Comments (discussion and conclusions):
Discussion
In general, I find the discussion works better than the results section. However, parts of the text is too close to results ( i.e. phrases such as ” those who responded “no” ..more frequently reported”. A lot of the information that is taken up in the discussion could be utilised to strengthen the background/rationale for the study. For example p 13, line 240-241 regarding childrens age.
Conclusion
Only a few sentences actually build on the findings while most is practical implications. The results do not warrant that family-friendly public policy work must be iterative etc. The conclusion needs to be based directly on the findings of the actual study. A solution could be to add a brief section of implications for research and practice just before the conclusion.
Authors’ response:
With regards to your comments relating to discussion and conclusion:
Discussion: We have rewritten parts of the discussion to make the text less repetitive. In addition to the overview of risk and support factors, we have strengthened the introduction with information previously presented in the discussion, such as the notion of cross-domain influences (page 2, lines 71–74). Regarding the statement of children’s age, please see page 3, lines 104–110 in the introduction.
Conclusion: You are correct. We have revised accordingly. See page 17, lines 383–391 (where we have added a brief section of implications for research and practice just before 5. Conclusions). Please also note that we have reformulated the conclusions and added a sentence stating the study contribution to section 5. Conclusions (page 17, lines 398–416).

Reviewer 2 Report
Thank you for the opportunity to read this interesting paper. Please see attached comments.

Author Response
Please see the attachment if you prefer to read authors' comments in PDF-format.
Authors’ responses to reviewer comments: reviewer 2:
Dear Reviewer,
Thank you for your input. We are glad to see that you find this manuscript interesting and well written. In the review report, you raise two minor points that we have addressed in the revised version. Please see our point-by-point response below.
Comment:
- In Table 1, it is not clear what education is classed as high or low educational level. I think this is solely due to the presentation of the material. For example, is “General upper secondary school” classed as high or low?
Authors’ response:
Thank you for this remark on the presentation of the material. Please note that in the revised version, Table 1 constitutes Table S1. We have revised the table to clarify the recoding of original categories. For example, as is now evident from Table S1, “general upper secondary school” is classified as low educational level.
Comment:
- Where responses to the items about Superior support and Co-worker support recoded? From Table 3 it appears not, but this isn’t clear in Table 1.
Authors’ response:
Thank you for noticing this. As noted in our response above, we think the initial presentation of the material led to this confusion. We have revised and hope it is now clear to the reader that responses to items related to superior support and co-worker support are kept in their original form. Please note that all variables which were kept in their original form in the analyses are marked with an asterisk (*) in Table S1 (previously Table 1).

Reviewer 3 Report
Dear authors,
I enjoyed reading your paper "Reconciling Work and Family Demands and Related Psycho-social Risk and Support Factors among Working Families: A Finnish National Survey Study". I really like your reasoning that it is especially important to investigate WIF/FIW among Finish families despite or maybe even exactly because of the comprehensive welfare state policies! Moreover, I think that it is an asset of your work that you consider not only both WIF and FIW, but also work- and family psychosocial factors at the same time!
I have two (minor) comments that I would like you to consider when revising your manuscript:
1. A lot of the literature you used on WIF/FIW is comparably old. There has been a large body of valuable literature, on family as well as on work-related risk factors, that I would recommend to consider including in your research.
2. My more important comment, however, concernc the dichotomization of your depend variables. While I like the idea of putting those without conflict in the focus, it does not become sufficiently clear to me why you did not exploit the full range of the conflict variables. Isn't it possible that differences in the extent of WIF/FIW are more important than just the differentiation of having no WIF/FIW at all vs. having WIF/FIW? Especially with regard to the Finish context I am wondering if you somewhat underestimate what the comprehensive wellfare state policies can not cover if you do not look at more "qualitative" differences in those measures. If you want to stick to your dichotomized measurement, I would recommend to add a stronger reasoning for that. The methodological arguments, however, are very convincing.
Author Response
Please see the attachment if you prefer to read authors' responses in PDF-format.
Authors’ responses to reviewer comments: reviewer 3
Dear Reviewer,
Thank you for your valuable comments. We are happy to see that you enjoyed reading the paper and that you agree with our reasoning concerning why this topic is important to examine in the Finnish welfare context. In the review report, you have provided two minor comments that we have considered when revising the paper. Please also see our point-by-point response below.
Comment:
- A lot of the literature you used on WIF/FIW is comparably old. There has been a large body of valuable literature, on family as well as on work-related risk factors, that I would recommend to consider including in your research.
Authors’ response:
Thank you for this important remark. Since our ambition was to keep the text concise, yet comprehensive (in accordance with journal guidelines), we have developed the introduction, focusing on key systematic review and meta-analytical findings in the revised and developed literature overview, also including newer references.
Comment:
- My more important comment, however, concernc the dichotomization of your depend variables. While I like the idea of putting those without conflict in the focus, it does not become sufficiently clear to me why you did not exploit the full range of the conflict variables. Isn't it possible that differences in the extent of WIF/FIW are more important than just the differentiation of having no WIF/FIW at all vs. having WIF/FIW? Especially with regard to the Finish context I am wondering if you somewhat underestimate what the comprehensive wellfare state policies can not cover if you do not look at more "qualitative" differences in those measures. If you want to stick to your dichotomized measurement, I would recommend to add a stronger reasoning for that. The methodological arguments, however, are very convincing.
Authors’ response:
Thank you for this reflective comment. While we agree with you stating that it is possible for within-group differences to exist with regards to the group of respondents reporting interference, this study focused on those reporting no interference by comparing these with those who reported interference. Since you approve of this focus, we have decided to keep the dependent variables dichotomized. However, we regard the added sentence in which we provide stronger reasoning for this decision, as suggested, as clarifying why we focused on this group of respondents. Please see tracked amendments on page 3, lines 124–126.

Round 2
Reviewer 1 Report
Congratulations to the authors for a well conducted revision of the manuscript. The positioning of the paper has improved considerably and the revised presenting of results makes it easier to get an overview. I find my comments and suggestions have been met appropriately and have no further to add.